# CIGMO: Categorical invariant representations in a deep generative framework

**Haruo Hosoya**[1]

[1]Brain Labs., ATR International, Kyoto, Japan

## Abstract

Data of general object images have two most common structures: (1) each object of a given shape can be rendered in multiple different views, and (2) shapes of objects can be categorized in such a way that the diversity of shapes is much larger across categories than within a category. Existing deep generative models can typically capture either structure, but not both. In this work, we introduce a novel deep generative model, called CIGMO, that can learn to represent category, shape, and view factors from image data. The model is comprised of multiple modules of shape representations that are each specialized to a particular category and disentangled from view representation, and can be learned using a group-based weakly supervised learning method. By empirical investigation, we show that our model can effectively discover categories of object shapes despite large view variation and quantitatively supersede various previous methods including the state-of-the-art invariant clustering algorithm. Further, we show that our approach using category-specialization can enhance the learned shape representation to better perform down-stream tasks such as one-shot object identification as well as shape-view disentanglement.

## 1 INTRODUCTION

In everyday life, we see objects in a great variety. Categories of objects are numerous and their shape variations are tremendously rich; different views make an object look totally different (Figure 1). Recent neuroscientific studies document how the primate visual system represents such complex objects in a characteristic, modular architecture, which is comprised of multiple cortical regions that each encode invariant features specialized to a particular object

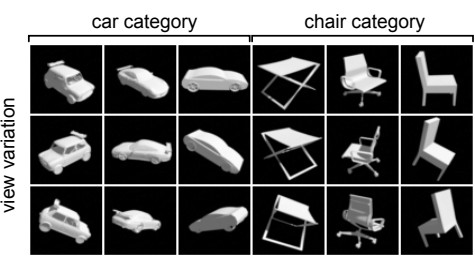

Figure 1: Examples of general object images. These include two categories (car and chair) each with three shape variations. The object of each shape is rendered in three different views.

category, such as faces [Freiwald and Tsao, 2010], body parts [Kumar et al., 2017], and other general categories [Srihasam et al., 2014, Bao et al., 2020]. Our work here takes inspiration from these biological findings for developing a novel learning model with "categorical invariant representation."

To be more specific, consider two most common domain structures of general object images. First, each object has a specific shape and can be rendered in multiple different views (e.g., 3D orientation) independently of shape. Second, the shapes of objects can be categorized in such a way that the diversity of shapes is much larger across categories than within a category. For example, Figure 1 illustrates various object images of car and chair categories, in which the shape of a particular car type is relatively similar to the shape of another car type but is substantially different from any chair type. Existing deep generative models typically capture either structure. The first structure on view variation is often handled by disentangling deep models, i.e., representation learning of mutually invariant factors of variation in the input [Bengio et al., 2013, Higgins et al., 2016, Chen et al., 2016, Bouchacourt et al., 2018, Hosoya, 2019, Mathieu et al., 2016, etc.], while the second structure on

*Accepted for the 38th Conference on Uncertainty in Artificial Intelligence* (UAI 2022).

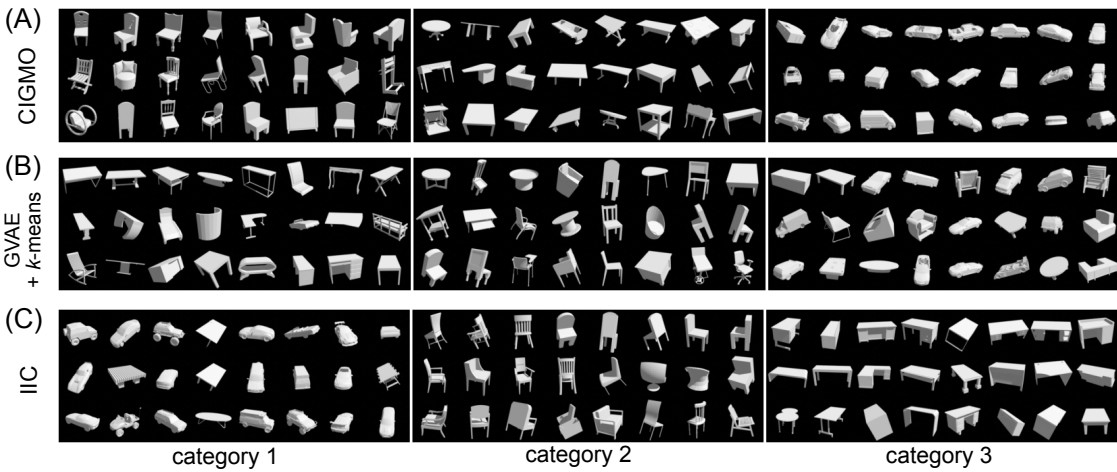

Figure 2: Examples of invariant clustering from (A) CIGMO, (B) GVAE [Hosoya, 2019] with $k$-means, and (C) IIC [Ji et al., 2019], for ShapeNet, in the case of 3 categories. Random 24 test images belonging to each estimated category are shown in a box. Note that the categories almost perfectly correspond to the chair, table, car image classes in (A). Such correspondence is much less clear in (B) and (C); in particular, cars are mixed with many other objects (category 3 in (B) and category 1 in (C)).

categorization can be handled by deep clustering methods [Jiang et al., 2017, Ji et al., 2019]. However, models that capture both structures in a single generative framework have rarely been studied.

In this work, to exploit both domain structures, we develop a probabilistic deep generative model that learns to encode three latent factors, namely, (1) category, (2) shape, and (3) view, from a dataset of general object images. However, this goal is generally difficult in a completely unsupervised setting, thus requiring some kind of "inductive bias" to be imposed [Locatello et al., 2020a]. To this end, we start with recently emerging, group-based weakly supervised learning [Mathieu et al., 2016, Bouchacourt et al., 2018, Chen et al., 2018, Hosoya, 2019], which can learn separate representations of shape and view from object images using no explicit labels, but only grouping information that links together different views of the same object. We extend this approach by introducing multiple modules of shape representations and then devising mechanisms to specialize each shape representation to a particular object category while disentangling it from view representation. We call the resulting model CIGMO (Categorical Invariant Generative MOdel).

We have empirically investigated representational advantages of our model. First, we found that our model allows for effectively solving the invariant clustering problem, that is, it can discover categories of object shapes in an unsupervised manner despite significant variation of object views. Our model can quantitatively outperform various previous methods including the state-of-the-art invariant clustering method [Ji et al., 2019] as well as combination of existing disentangling and clustering methods; Figure 2 shows demonstrat-

ing examples. Second, we found that our approach using category-specialization can enhance the learned shape representation to perform better multiple tasks, including one-shot identification (object recognition given one example per shape) as well as shape-view disentanglement in multiple criteria. Thus, we propose CIGMO as a novel learning approach to represent object images in a general manner using category, shape, and view latent variables, which can provide more precise information for down-stream tasks than typical approaches to represent only part of those variables. The source code written in `pytorch` is available at `https://github.com/HaruoHosoya/cigmo`.

## 2 RELATED WORK

A number of studies exist for disentangling deep generative models. A particularly relevant technique is the group-based disentangling approach that can learn to separate content (shape) and view variables from grouped data [Mathieu et al., 2016, Bouchacourt et al., 2018, Chen et al., 2018, Hosoya, 2019, Locatello et al., 2020b]. However, these studies most often use image datasets of a single object category, paying no attention to the large cross-category diversity of object shapes mentioned in the introduction. Our model can be seen as a generalization of this approach with categorization, where our specific contribution is how to handle grouped data in the presence of multiple categories (Section 3.2); such issue would not arise in a non-group-based setting, e.g., [Kingma et al., 2014].

Although our focus here is weak supervision, we should mention that there are various approaches to use explicit

labels for enhancing disentangling performance, such as semi-supervised learning [Kingma et al., 2014, Siddharth et al., 2017] or adversarial learning to promote disentanglement [Lample et al., 2017, Mathieu et al., 2016]. Also, as group-based learning was originally inspired by temporal coherence principle [Földiák, 1991] (i.e., the object identity is often stable over time), some weakly supervised disentangling approaches have explicitly used it [Yang et al., 2015].

Some studies have proposed unsupervised disentangling algorithms that impose statistical constraints on the latent representation such as information maximization [Higgins et al., 2016, Chen et al., 2016, Dupont, 2018]. However, we emphasize that the problem that they try to solve is very different from ours. That is, their models can learn arbitrarily many continuous variables that are each single-dimensional, whereas our model (as well as all other disentangling models cited so far) can learn exactly two continuous variables that are each multi-dimensional (shape and view).

Our study is also related to recent deep clustering methods. In particular, a latest approach proposes group-based learning for invariant clustering, which maximizes mutual information between categorical posterior distributions for paired images [Ji et al., 2019]. This method has exhibited remarkable performance on natural images under various view variation; Section 4 gives an empirical comparison with our method. Although earlier work combines variational autoencoders (VAE) [Kingma and Welling, 2014] with a conventional clustering method (e.g., Gaussian mixture), such approach seems to be limited in capturing large object view variation [Jiang et al., 2017]. In any case, these methods are specialized to clustering and throw away all information other than the category.

The group-based learning is loosely related to contrastive learning [Jaiswal et al., 2020], which is a self-supervised learning approach to make representations of "positive" data pairs near (e.g., objects of the same shape) and those of "negative pairs" distant (e.g., objects of different shapes). Such negative data (not used in our study) could help drastically improve performance, though it incurs substantial technical complication (large batch size, memory bank, etc.).

# 3 CIGMO: CATEGORICAL INVARIANT GENERATIVE MODEL

## 3.1 MODEL

In our model construction, similarly to the previous group-based disentangling approaches [Mathieu et al., 2016, Bouchacourt et al., 2018, Hosoya, 2019, Chen et al., 2018, Locatello et al., 2020b], we assume a dataset that groups together multiple object images of the same shape but possibly in different views. Here, shape refers to the property of object that is invariant in view, where view is defined depend-

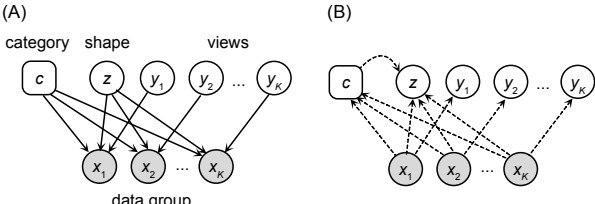

Figure 3: (A) The graphical model. Each instance $x_k$ in a data group is generated from a category $c$, a shape $z$, and a view $y_k$. Round boxes are discrete variables; circles are continuous variables; shaded are visible variables. (B) The inference flow. Each hidden variable is inferred from the set of incoming variables.

ing on the dataset. For example, 3-dimensional viewpoints are considered as views in the object images in Figure 1 (columns). Given such dataset, we can extract the shape as a group-common factor and the view as an instance-specific factor. In this study here, we generalize this idea with category-modular representation. That is, we assume multiple object categories each of which includes a distinct and potentially infinite set of shapes. Since the variety of shapes can be category-specific (e.g., shapes of chairs vary in a different way from those of cars), we endow each category with its own specialized representation (i.e., separate encoders and decoders).

Formally, we assume a grouped dataset $\mathbb{D} = \{(\boldsymbol{x}_1^{(n)}, \ldots, \boldsymbol{x}_K^{(n)}) \mid \boldsymbol{x}_k^{(n)} \in \mathbb{R}^D, n = 1, \ldots, N\}$, where each data point is a group (tuple) of $K$ data instances (e.g., images); we assume that the groups are i.i.d. For a data group $(\boldsymbol{x}_1, \ldots, \boldsymbol{x}_K)$, we consider three types of hidden variables: category $c \in \{1, \ldots, C\}$, shape $\boldsymbol{z} \in \mathbb{R}^M$, and views $\boldsymbol{y}_1, \ldots, \boldsymbol{y}_K \in \mathbb{R}^L$ (omitting the superscript $(n)$), where the category and shape are common for the group while the views are specific to each instance. We consider the following generative model (Figure 3(A)):

$$p(c) = \pi_c$$
$$p(\boldsymbol{z}) = \mathcal{N}_M(0, \boldsymbol{I})$$
$$p(\boldsymbol{y}_k) = \mathcal{N}_L(0, \boldsymbol{I})$$
$$p(\boldsymbol{x}_k | \boldsymbol{y}_k, \boldsymbol{z}, c) = \mathcal{N}_D(f_c(\boldsymbol{y}_k, \boldsymbol{z}), \boldsymbol{I})$$

for $c = 1, \ldots, C$ and $k = 1, \ldots, K$. Here, $\pi_c$ is a category prior with $\sum_{c=1}^C \pi_c = 1$ and $f_c$ is a decoder deep net defined for category $c$ corresponding to the category-specific module of shape representation. In the generative process, the category $c$ is first drawn from the categorical distribution $(\pi_1, \ldots, \pi_C)$, while the shape $\boldsymbol{z}$ and views $\boldsymbol{y}_k$ are drawn from standard Gaussian priors. Then, each data instance $\boldsymbol{x}_k$ is generated by the decoder $f_c$ for the selected category $c$, which is applied to the group-common shape $\boldsymbol{z}$ and the instance-specific view $\boldsymbol{y}_k$ (added with Gaussian noise of unit variance). In other words, different data instances for

a group are generated from the same shape and different views.

## 3.2 LEARNING

Having defined a generative model as above, we expect category, shape, and view representations to arise as latent variables after fitting the model to a grouped dataset. To derive a concrete algorithm, we use variational autoencoders [Kingma and Welling, 2014] as a basic methodology, where we develop a specific architecture to solve our particular problem.

As the most important step, we specify inference models to encode approximate posterior distributions (Figure 3(B)). First, we estimate the posterior probability for category $c$ as follows:

$$q(c|\boldsymbol{x}_1,\ldots,\boldsymbol{x}_K) = \frac{1}{K}\sum_{k=1}^{K} u^{(c)}(\boldsymbol{x}_k) \qquad (1)$$

Here, $u$ is a categorizer deep net that computes, for an individual instance $\boldsymbol{x}_k$, a probability distribution over the categories ($\sum_{c=1}^{C} u^{(c)}(\boldsymbol{x}_k) = 1$) [Kingma et al., 2014]. Since we have $K$ such instances, we take the average over the instance-specific distributions to obtain the group-common distribution. The averaging operation is justified as estimation of the expected probability of each category within the given group.

Next, we infer the posterior for each instance-specific view $\boldsymbol{y}_k$ from the input $\boldsymbol{x}_k$ as follows:

$$q(\boldsymbol{y}_k|\boldsymbol{x}_k) = \mathcal{N}_L\left(g(\boldsymbol{x}_k), \mathrm{diag}(r(\boldsymbol{x}_k))\right) \qquad (2)$$

where $g$ and $r$ are encoder deep nets to specify the mean and variance, respectively. Here, we use the view representation that does not depend on the category, assuming a "universal" view space. Then, we estimate the posterior for group-common shape $\boldsymbol{z}$ from inputs $\boldsymbol{x}_1,\ldots,\boldsymbol{x}_K$ as follows:

$$q(\boldsymbol{z}|\boldsymbol{x}_1,\ldots,\boldsymbol{x}_K,c) =$$
$$\mathcal{N}_M\left(\frac{1}{K}\sum_{k=1}^{K} h_c(\boldsymbol{x}_k), \frac{1}{K}\sum_{k=1}^{K}\mathrm{diag}(s_c(\boldsymbol{x}_k))\right) \qquad (3)$$

This time, shape representation does depend on the category, unlike views. Thus, the encoder deep nets $h_c$ and $s_c$ are defined for each category $c$, which compute the mean and variance, respectively, for each individual shape for $\boldsymbol{x}_k$. We then obtain the group-common shape $\boldsymbol{z}$ as the average over all the individual shapes [Hosoya, 2019].

For training, we define the following variational lower bound of the marginal log likelihood for a data point:

$$\mathcal{L}(\phi;\vec{\boldsymbol{x}}) = \mathcal{L}_{\mathrm{recon}} + \mathcal{L}_{\mathrm{KL}} \text{ with}$$

$$\mathcal{L}_{\mathrm{recon}} = \mathbb{E}_{q(\vec{\boldsymbol{y}},\boldsymbol{z},c|\vec{\boldsymbol{x}})}\left[\sum_{k=1}^{K}\log p(\boldsymbol{x}_k|\boldsymbol{y}_k,\boldsymbol{z},c)\right]$$

$$\mathcal{L}_{\mathrm{KL}} = -D_{\mathrm{KL}}(q(\vec{\boldsymbol{y}},\boldsymbol{z},c|\vec{\boldsymbol{x}})\|p(\vec{\boldsymbol{y}},\boldsymbol{z},c))$$

where $\vec{\boldsymbol{x}}$ stands for $(\boldsymbol{x}_1,\ldots,\boldsymbol{x}_K)$, etc., and $\phi$ is the set of all weight parameters in the categorizer, encoder, and decoder deep nets. We compute the reconstruction term $\mathcal{L}_{\mathrm{recon}}$ as follows:

$$\mathcal{L}_{\mathrm{recon}} = \sum_{c=1}^{C} q(c|\vec{\boldsymbol{x}})\mathbb{E}_{q(\vec{\boldsymbol{y}},\boldsymbol{z}|\vec{\boldsymbol{x}},c)}\left[\sum_{k=1}^{K}\log p(\boldsymbol{x}_k|\boldsymbol{y}_k,\boldsymbol{z},c)\right]$$

$$\approx \sum_{c=1}^{C} q(c|\vec{\boldsymbol{x}})\sum_{k=1}^{K}\log p(\boldsymbol{x}_k|\boldsymbol{y}_k,\boldsymbol{z},c)$$

where we approximate the expectation using one sample $\boldsymbol{z} \sim q(\boldsymbol{z}|\vec{\boldsymbol{x}})$ and $\boldsymbol{y}_k \sim q(\boldsymbol{y}_k|\boldsymbol{x}_k,c)$ for each $k$, but directly use the probability value $q(c|\vec{\boldsymbol{x}})$ for $c$. The latter is crucial for making the loss function differentiable. The KL term $\mathcal{L}_{\mathrm{KL}}$ is computed as follows:

$$\mathcal{L}_{\mathrm{KL}} = -D_{\mathrm{KL}}(q(c|\vec{\boldsymbol{x}})\|p(c))$$
$$-\sum_{k=1}^{K} D_{\mathrm{KL}}(q(\boldsymbol{y}_k|\boldsymbol{x}_k)\|p(\boldsymbol{y}_k))$$
$$-\sum_{c=1}^{C} q(c|\vec{\boldsymbol{x}})D_{\mathrm{KL}}(q(\boldsymbol{z}|\vec{\boldsymbol{x}},c)\|p(\boldsymbol{z}))$$

Finally, our training procedure is to maximize the lower bound for the entire dataset with respect to the weight parameters: $\hat{\phi} = \arg\max_{\phi}\frac{1}{N}\sum_{n=1}^{N}\mathcal{L}(\phi;\boldsymbol{x}_1^{(n)},\ldots,\boldsymbol{x}_K^{(n)})$.

Figure 4 depicts the outline of our learning algorithm. We emphasize here that, even though the architecture has rather intertwined interaction between grouping and categorization, our model can be formalized elegantly in a probabilistic generative framework, can be trained in an end-to-end manner only using a grouped dataset and no explicit label for category and so on (nor any pre-trained model), and can perform well for multiple down-stream tasks (Section 4).

## 3.3 MODEL VARIANTS

The above construction is only one approach to categorical invariant generative models. There can indeed be a number of variants. First, instead of combining categorical distributions by averaging as in equation 1, we could take the (normalized) product: $q(c|\boldsymbol{x}_1,\ldots,\boldsymbol{x}_K) \propto \prod_{k=1}^{K} u^{(c)}(\boldsymbol{x}_k)$ (which can be interpreted as evidence accumulation); or taking the softmax on the averaged logits: $q(c|\boldsymbol{x}_1,\ldots,\boldsymbol{x}_K) \propto \exp\left(\frac{1}{K}\sum_{k=1}^{K}\tilde{u}^{(c)}(\boldsymbol{x}_k)\right)$ where $\tilde{u}^{(c)}$ is the logit of $u^{(c)}$ (which estimates expectation at the level of neural network

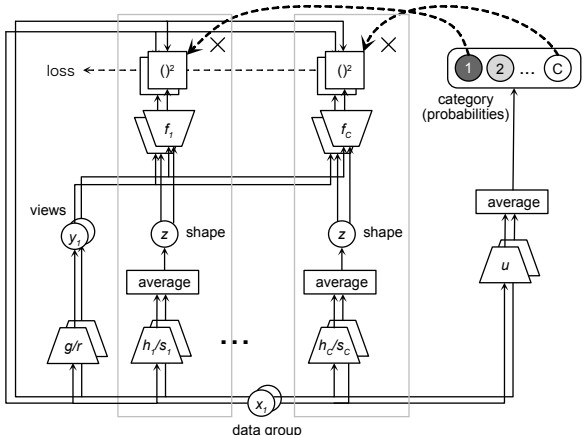

Figure 4: A diagrammatic outline of CIGMO learning algorithm. The entire workflow consists of three kinds of networks corresponding to views (left-most), category (right-most), and shapes in $C$ modules. Given an input group of instances $x_k$ (bottom), an instance-specific view $y_k$ is computed by encoders $g$ and $r$. In each module $c$, a group-common shape $z$ is computed by encoders $h_c$ and $s_c$ followed by averaging. Then, new data instances generated by decoder $f_c$ from each shape and view are compared with the original data for obtaining the reconstruction error (loss). This process is repeated for all modules. In parallel, the posterior probability for category $c$ is computed by the categorizer $u$ followed by averaging and multiplied with the reconstruction error for the corresponding module. Note that other probabilistic mechanisms (e.g., priors) are omitted here for brevity.

outputs). Second, instead of the "universal" view space as in equation 2, we could consider a category-dependent view representation: $q(\boldsymbol{y}_k|\boldsymbol{x}_k, c) = \mathcal{N}_L\left(g_c(\boldsymbol{x}_k), \operatorname{diag}(r_c(\boldsymbol{x}_k))\right)$ using encoders $g_c$ and $r_c$ defined for each category $c$. This is sensible since, e.g., frontal views for chairs could arguably mean differently from frontal views for tables. For these two, Supplementary Materials make empirical comparison, showing that our original approach in Section 3.2 gives slightly better performance than the variants.

In addition, we can think of a relatively different approach that takes an existing group-based disentangling model [Bouchacourt et al., 2018, Hosoya, 2019, Locatello et al., 2020b], but adds Gaussian mixture prior on the shape variable. Formally, assume

$$p(c) = \pi_c$$
$$p(\boldsymbol{z}'|c) = \mathcal{N}_M(\boldsymbol{b}_c, \boldsymbol{A}_c)$$
$$p(\boldsymbol{y}_k) = \mathcal{N}_L(0, \boldsymbol{I})$$
$$p(\boldsymbol{x}_k|\boldsymbol{y}_k, \boldsymbol{z}') = \mathcal{N}_D(f(\boldsymbol{y}_k, \boldsymbol{z}'), \boldsymbol{I})$$

where $\boldsymbol{b}_c$ and $\boldsymbol{A}_c$ are a mean vector and covariance ma-

trix, respectively, corresponding to category $c$; $f$ is a deep net decoder (not depending on the category). However, the following can be proved.

**Theorem 1.** *Group-based disentangling models with Gaussian mixture prior are a special case of* CIGMO.

*Proof.* The result immediately follows by letting $\boldsymbol{z} = \boldsymbol{A}_c^{-\frac{1}{2}}(\boldsymbol{z}' - \boldsymbol{b}_c)$ and $f_c(\boldsymbol{y}_k, \boldsymbol{z}) = f(\boldsymbol{y}_k, \boldsymbol{A}_c^{\frac{1}{2}}\boldsymbol{z} + \boldsymbol{b}_c)$. $\square$

This corresponds to the specific architecture in CIGMO that shares the part of decoder networks after the first fully connected layer. In fact, we precisely use such architecture in the experiments (Supplementary Materials).

## 4  EXPERIMENTS

We have applied CIGMO as described in Section 3 to two image datasets: ShapeNet (general objects) and MVC Cloth (cloth images). Below, we outline the experimental set-up and show the results.

### 4.1  SHAPENET

For the first set of experiment, we derived a dataset of multi-viewed object images from 3D models in ShapeNet database [Chang et al., 2015]. The dataset consisted of 10 pre-defined object classes: car, chair, table, airplane, lamp, boat, box, display, truck, and vase. Within each class, we had a large number of objects with specific shapes, which we distinguished by object identities. We rendered each object in 30 views (3-dimensional viewpoints) in a single lighting condition. We split the training and test sets, which consisted of 21888 and 6210 object identities, respectively. (See Supplementary Materials for more details.) We also created subset versions with 3 or 5 object classes. For training data, we formed groups of images of the same object in random 3 views ($K = 3$), though our approach works for any choice of group size of 2 or larger (Section 4.1.4) We used object identity labels (not class labels) for grouping. Importantly, however, after this step, we never used any label during training.

To train a CIGMO model, we used the following setting. We set the number of categories in the model either to the number of classes in the data ($C = 3, 5,$ or $10$), or to a larger number, depending on the task. We set the shape dimension $M = 100$ and the view dimension $L = 3$ (where a very low view dimension was used to avoid the view variable taking over all the input information and thereby the shape variable becoming degenerate). For the categorizer, encoder, and decoder deep nets, we adopted commonly used architectures of convolutional or deconvolutional neural networks. Since the model had so many deep nets, a large part of the networks was shared to save the memory space. For simplicity,

Table 1: Invariant clustering accuracy (%) for ShapeNet. The mean and SD over 10 model instances are shown; the best method with the highest mean score is shown in the bold font; the ∗-mark indicates statistical significance relative to the best method (t-test; $p < 0.05$); ditto for the remaining tables.

| Methods | 3 classes | 5 classes | 10 classes |
|---|---|---|---|
| chance level | 33.33 | 20.00 | 10.00 |
| IIC | $85.25 \pm 13.74$ | $81.10 \pm 7.33^*$ | $60.84 \pm 1.45^*$ |
| IIC (overclust.) | $79.86 \pm 13.78^*$ | $81.87 \pm 4.57^*$ | $59.73 \pm 1.49^*$ |
| VAE + $k$-means | $66.41 \pm 5.69^*$ | $50.83 \pm 3.85^*$ | $37.07 \pm 1.00^*$ |
| Mixture of VAEs | $82.35 \pm 5.66^*$ | $65.73 \pm 6.24^*$ | $40.86 \pm 3.58^*$ |
| MLVAE + $k$-means | $82.04 \pm 7.78^*$ | $70.68 \pm 5.04^*$ | $54.47 \pm 1.92^*$ |
| GVAE + $k$-means | $73.20 \pm 10.93^*$ | $69.42 \pm 3.47^*$ | $52.55 \pm 2.74^*$ |
| **CIGMO** | $\mathbf{94.83} \pm 6.06$ | $\mathbf{89.36} \pm 4.53$ | $\mathbf{68.53} \pm 4.24$ |

Table 2: One-shot identification accuracy (%) for ShapeNet.

| Methods | 3 classes (3705 objs.) | 5 classes (4977 objs.) | 10 classes (6210 objs.) |
|---|---|---|---|
| chance level | 0.03 | 0.02 | 0.02 |
| VAE | $2.17 \pm 0.03^*$ | $3.49 \pm 0.04^*$ | $3.43 \pm 0.02^*$ |
| Mixture of VAEs | $2.31 \pm 0.04^*$ | $3.71 \pm 0.06^*$ | $3.55 \pm 0.06^*$ |
| MLVAE | $24.00 \pm 0.43^*$ | $20.30 \pm 0.26^*$ | $17.93 \pm 0.29^*$ |
| GVAE | $24.51 \pm 0.44^*$ | $20.30 \pm 0.24^*$ | $17.91 \pm 0.20^*$ |
| **CIGMO** | $\mathbf{27.33} \pm 0.55$ | $\mathbf{24.51} \pm 0.68$ | $\mathbf{21.79} \pm 0.71$ |

we fixed the category prior $\pi_c = 1/C$. Supplementary Materials give more details on the architecture. For optimization, we used Adam [Kingma and Ba, 2015] with mini-batches of size 100 and ran 20 epochs.

We evaluated the trained models using test data, which were ungrouped and contained objects of the same classes as training data but of different identities. Below, we describe the evaluation in three parts: (1) invariant clustering task, (2) one-shot identification task, and (3) category-wise shape-view disentangling. Supplementary Materials give comparison of the model variants discussed in Section 3.3.

### 4.1.1 Invariant clustering

The goal of this task is to perform clustering of input images regardless of the view. Since our learning method has already estimated the latent category variable without using labels, the rest is to simply infer the most probable category from a given test image, $\hat{c} = \arg\max_c q(c|\boldsymbol{x})$.

For comparison, we considered the following baseline models. First, we included two group-based disentangling methods, namely, Group-based VAE (GVAE) [Hosoya, 2019] and Multi-Level VAE (MLVAE) [Bouchacourt et al., 2018]. These can learn to separately infer shape and view from object images; we performed $k$-means clustering on the learned shape variable. In addition, we incorporated In-

variant Information Clustering (IIC) [Ji et al., 2019], the state-of-the-art, group-based method specialized to invariant clustering. IIC came in two versions: with and without regularization using 5-times overclustering. For all the baseline methods above, we gave exactly the same grouped dataset for training. In addition to these, we examined, as an ablation study, two non-group-based methods: mixture of VAEs and vanilla VAE with $k$-means.

Figure 2(A) shows an example result from a CIGMO model with 3 categories applied to the 3-class dataset, where each box shows random test images belonging to each estimated category. This demonstrates a very precise clustering of objects achieved by our model, which is quite remarkable given the large view variation and no category label used during training. Figures 2(B) and (C) show analogous examples from GVAE with $k$-means as well as IIC, clearly indicating lower performance compared to CIGMO.

For quantitative comparison, we measured the performance of invariant clustering in two standard criteria. The first is classification accuracy. For this, since we needed to compute the best category-to-class mapping by the Hungarian algorithm [Munkres, 1957] and since this step requires the number of categories in the model to equal the number of classes in the data, we used only such models corresponding to each dataset with 3, 5, or 10 classes. Table 1 summarizes the results (for 10 model instances for each method). Generally, CIGMO outperformed the other methods in all

Table 3: Degree of shape-view disentanglement for ShapeNet by three measures. Left: swapping error (lower is better); middle: neural network classification accuracy (%) for object identity from the shape (higher is better); right: that for view variable (lower is better); weighted average over categories.

| Methods | swapping error | | shape → id (%) | | view → id (%) | |
|---|---|---|---|---|---|---|
| | 5 classes | 10 classes | 5 classes | 10 classes | 5 classes | 10 classes |
| MLVAE | $0.497 \pm 0.005^*$ | $0.554 \pm 0.010^*$ | $47.30 \pm 0.78^*$ | $42.96 \pm 0.74^*$ | $0.60 \pm 0.05$ | $0.52 \pm 0.08$ |
| GVAE | $0.491 \pm 0.004^*$ | $0.549 \pm 0.005^*$ | $48.34 \pm 0.55^*$ | $43.87 \pm 0.30^*$ | $\mathbf{0.56} \pm 0.05$ | $\mathbf{0.48} \pm 0.04$ |
| **CIGMO** | $\mathbf{0.300} \pm 0.025$ | $\mathbf{0.340} \pm 0.035$ | $\mathbf{50.91} \pm 0.97$ | $\mathbf{46.28} \pm 0.87$ | $0.65 \pm 0.05^*$ | $0.67 \pm 0.07^*$ |

cases with a large margin (mostly with statistical significance). More specifically, first, CIGMO performed better than GVAE or MLVAE with $k$-means, which shows that category information can be more clearly represented with multiple category-specific shape representations, compared to a general shape representation with a conventional clustering method. Second, CIGMO superseded IIC, which shows that general modeling of object images with category, shape, and view latent variables can do better than directly solving the specific task. This was the case both with and without overclustering; in fact, we could not find a consistent improvement by overclustering in IIC, contrary to the claim by Ji et al. [2019]. Third, CIGMO showed significantly higher scores than mixture of VAEs, which confirms the large impact of group-based learning. Taken together, these results demonstrate the efficacy of the categorical invariant representation in the invariant clustering task.

The second criterion is Adjusted Rand Index (ARI), which provides a similarity measure between sample-to-category and sample-to-class assignments [Hubert and Arabie, 1985]. Since this criterion allows any number of categories in the model, we assessed the change of scores for different numbers of categories. Figure 5 shows the results for the 3-class and 10-class datasets. Generally, all methods decreased performance while the number of categories increased. However, CIGMO reasonably retained the score even for larger numbers of categories, while the other methods exhibited a sharp drop. This is because CIGMO tends to use just a necessary number of categories and leave the remaining categories as degenerate (to which no input belongs), whereas the other methods try to find out as many clusters as possible and thus always use all categories. This property of CIGMO would be useful in a realistic setting where the number of true categories is unknown *a priori*.

### 4.1.2 One-shot identification

The goal of this task is to perform object recognition given only one example per identity. More precisely, we randomly pick up a subset of the test images consisting of exactly one image for each object identity and then identify the objects of the remaining test images. Since the test objects

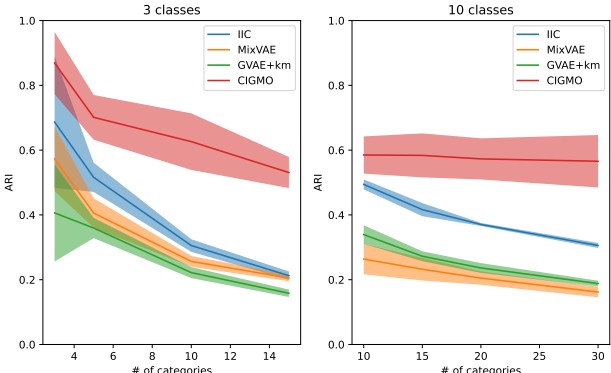

Figure 5: Adjusted Rand Index for ShapeNet while increasing the number of categories in the model (line: mean, shade: SD).

are disjoint from the training objects, we deal with only unseen objects for the model. This task can measure the strength of view-invariance of shape representation: if shape code is perfectly invariant in view, then all images of the same object should be mapped to an identical point in the shape space. Note that our purpose here is not to infer the class but the object identity, unlike invariant clustering.

Thus, we compared overall accuracy of one-shot identification for CIGMO and other models. For this, we performed a nearest-neighbor method according to Euclidean distance in the shape space. Here, the shape space was defined depending on the method. For GVAE or MLVAE, the shape variable $\boldsymbol{z} = h(\boldsymbol{x})$ directly defined the shape space. For CIGMO, since the shape representation depended on the category, we first defined $\boldsymbol{z}^c = h_c(\boldsymbol{x})$ for $c = \hat{c}$ (where $\hat{c}$ is inferred from the input) and $\boldsymbol{z}^c = 0$ otherwise, and then concatenated them together: $[\boldsymbol{z}^1, \ldots, \boldsymbol{z}^C]$. This gave us category-dependent shape vectors that could be directly compared. For VAE or mixture of VAEs, we used the entire latent variable in place of shape variable. For simplicity, we again used models with the same number of categories as the image classes.

Table 2 summarizes the results. Overall, CIGMO performed the best among the compared methods in all cases (with sta-

tistical significance). In particular, it performed significantly better than GVAE and MLVAE, which indicates that shapes can be represented more precisely with category specialization than without. In addition, our model outperformed, by far, mixture of VAEs, showing the successful disentanglement of shape from view. These results, again, indicate the advantage of the categorical and invariant representations in the one-shot identification task. (Note also that the scores were remarkably high even for up to 6210-way classification by one shot.)

### 4.1.3 Category-wise shape-view disentanglement

CIGMO provides category-specific shape representations that are disentangled from views. We quantified the degree of disentanglement in two ways. First, we generated a number of "swapping" images by the decoder from the view of one image and the shape of another, and calculated the (normalized) mean squared error between the generated images and the ground truths (swapping error). Second, we measured how much information the shape or view variable contained on object identity, for which we trained two-layer neural networks on either variable for classification (identity inference) [Mathieu et al., 2016, Bouchacourt et al., 2018]. A better disentangled representation would give a higher accuracy from the shape variable and a lower accuracy from the view variable. For each analysis, we conducted the measurement for each category using the belonging test images and took the average weighted by the number of those images. Table 3 summarizes the results. Overall, CIGMO tended to give better results than the baselines, except for object information in the view variable (though the compared accuracies were all negligibly low).

### 4.1.4 Group size variation

In Section 4.1, we have argued that our model works as long as the group size is two or larger. This is because the shape should be in principle the same no matter how many instances are in each group. However, a larger group size could more easily stabilize the model training, thus potentially changing the result quantitatively. Figure 6 shows the change of performance while the group size was varied. The result indeed shows that increasing the group size tends to slightly improve the performance (except for the saturation in invariant clustering for fewer classes).

### 4.2 MVC CLOTH

For the second set of experiment, we used a dataset of multi-viewed clothing images based on MVC Cloth dataset [Liu et al., 2016]. The dataset contains a number of photos of cloths worn by fashion models and taken in multiple viewpoints (Supplementary Materials). Unlike ShapeNet, this

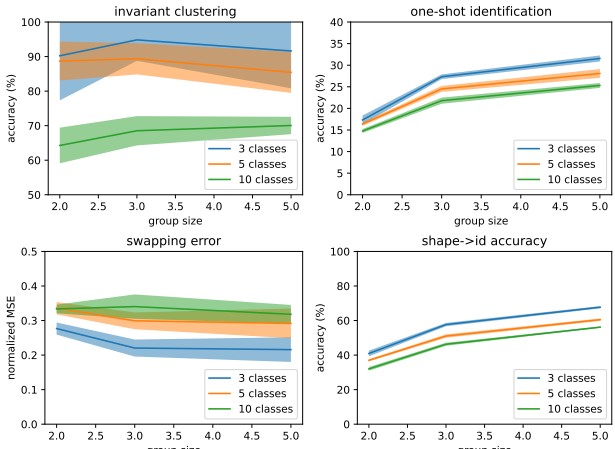

Figure 6: Comparison of performance in invariant clustering, one-shot identification, swapping, and shape-to-identity when the group size is varied.

dataset provides no class label. Therefore our focus here is to demonstrate what kind of unknown but interesting categories can be discovered by our model. We split the dataset into the training and test sets each consisting of ∼112K and ∼28K images with disjoint cloth types; the training images were grouped for the same cloth types (group size 3). For model training, we arbitrarily set the number of categories to 7 in a CIGMO. Other training conditions were identical as before.

Figure 7(A) illustrates an invariant clustering result, where only 4 categories were shown as the other 3 categories were degenerate. By inspection, category 1 represents whole-body cloths like suits and dresses; category 2 represents tops like sweaters, jackets, blouses, and shirts typically with long sleeves; category 3 represents bottoms like trousers and jeans; category 4 represents tops like shirts typically with short sleeves. To characterize the categories more systematically, we used the attribute information provided by the dataset (264 boolean attributes for each image; see Supplementary Materials). For each estimated category and each attribute label, we calculated F1-score to measure their relevance [Fawcett, 2006]. Table 4 gives top 10 most relevant attributes to each category; we can see good matching of the result with the visual impression from Figure 7(A).

Figure 7(B) shows a swapping result from the same CIGMO model, where each matrix corresponds to a category and shows the images generated from the shape of one image (left column) and the view of another (top row). We can see that the shape and view representations are well disentangled in each category, as the generated images are clearly aligned for the shape in rows and for the view in columns.[1]

---

[1]Note that our goal here is not to generate sharp images. Generally, it is well known that, compared to VAE-based methods, GAN-based methods tend to generate sharper but often more cor-

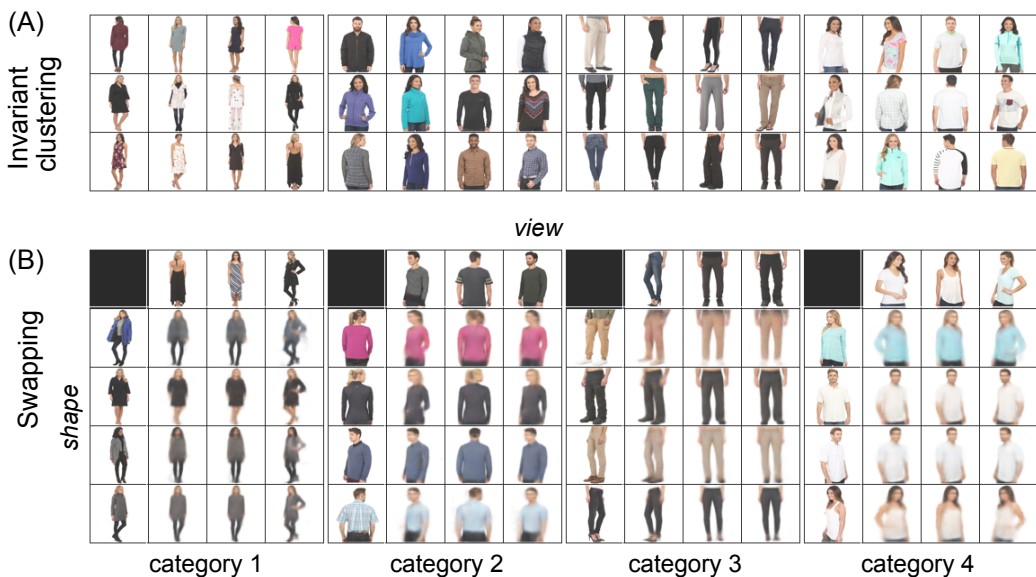

Figure 7: Results from a CIGMO model trained for MVC Cloth dataset. (A) Invariant clustering. Random images belonging to each estimated category are shown in a box. (B) Swapping. For each category, an image is generated from the view of one image in the top row and the shape of another image in the left column.

Table 4: Top 10 most relevant attributes to each category in the model in Figure 7. Each attribute name is shown with the F1-score in parentheses.

| Catg. | Attributes (F1-scores) |
|---|---|
| 1 | Short (0.45); Sleeveless (0.33); hundred2O (0.32); Polyester (0.32); AlineDresses (0.31); Sheath-Dresses (0.31); Black (0.29); hundred2U (0.28); KneeLength (0.24); Nylon (0.20); |
| 2 | hundred2U (0.65); LongSleeves (0.63); hundred1U (0.57); Polyester (0.55); Pullover (0.47); Cotton (0.47); TShirts (0.39); Black (0.38); fiftyU (0.31); Nylon (0.28); |
| 3 | Denim (0.49); Cotton (0.38); hundred2U (0.34); hundred1U (0.33); Polyester (0.27); Black (0.27); PullOn (0.27); Blue (0.27); ContrastStitching (0.26); Leggings (0.26); |
| 4 | White (0.57); ShortSleeves (0.27); Pullover (0.25); TShirts (0.24); fiftyU (0.19); hundred1U (0.19); Crew (0.19); ButtonUpShirts (0.17); hundred2U (0.17); Cotton (0.16); |

glement with views, which can learn to represent category, shape, and view latent variables with group-based weak supervision. We have shown empirical representational advantages that allow our model to outperform previous methods in multiple down-stream tasks such as invariant clustering and one-shot identification. One drawback is scalability: the per-step time complexity proportional to the number of categories, due to the reparametrization trick incompatible with category variables (Section 3.2). The Gumbel-Softmax technique is well-known for this type of problem [Jang et al., 2017], but did not work in our case for an unknown reason, though not so surprising since it is only a heuristics.

Future investigation may include improvements in overall task performance as well as image generation quality, possibly with the aid of adversarial learning [Goodfellow et al., 2014, Mathieu et al., 2016, Chen et al., 2018], and application to more realistic settings. Lastly, back to our original inspiration, we are keen to pursue the biological relationship of CIGMO to the primate higher visual cortex, continuing our previous investigations [Hosoya and Hyvärinen, 2017, Raman and Hosoya, 2020].

## 5 CONCLUSION

In this paper, taking inspiration from the primate higher vision, we have proposed CIGMO as a deep generative model that has category-modular shape representations in disentan-

---

rupted images especially when the number of training images is not so large, e.g., [Chen et al., 2018, Figs. 7 and 12].

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

**Acknowledgements**

This work has been supported by Commissioned Research of NICT (1940201) and Grants-in-Aid for Scientific Research (19H04999 and 21K19812).
