# OpenReview forum: "CIGMO: Categorical invariant representations in a deep generative framework"
_auai.org/UAI/2022/Conference — UAI 2022 Poster_

### Official Review · Reviewer_Mnhw · 2022-03-24

**Q2(1) Originality/Novelty:** 2
**Q2(2) Significance/Impact:** 2
**Q2(3) Correctness/Technical Quality:** 2
**Q2(6) Clarity Of Writing:** 2
**Q6 Overall Score:** 5
**Q8 Confidence In Your Score:** 2

**Q1 Summary And Contributions:**

This paper introduces a novel model, named CIGMO, to decouple category, shape and  view factors from image data. Experiments show this model can discover categories of object shapes disregarding large view variation. In addition, this model will benefit various down-stream tasks such as one-shot object identification and shape-view disentanglement.

**Q2 Assessment Of The Paper:**

More detailed information regarding each of these aspects is given below:

**Q2(4) Quality Of Experiments (Optional):**

2: Fair: The experimental evaluation is weak: important baselines are missing, or the results do not adequately support the main claims.

**Q2(5) Reproducibility:**

2: Fair: Key resources (e.g., proofs, code, data) are unavailable but key details (e.g., proof sketches, experimental setup) are sufficiently well-described for an expert to confidently reproduce the main results.

**Q3 Main Strengths:**

This paper contributes new ideas and this idea seems works according to authors experiments. I think proposed method will benefit representation learning.

The technique in this paper is sound.

The experiments in this paper are sufficient. Invariant clustering, one-shot identification and category-wise shape-view disentanglement are reported.


**Q4 Main Weakness:**

I think the section 3 is not well-written.

Section 3.1 discribes a probablity model. Section 3.2 should list the objective. The "learning" title likely mislead readers.  According to my understanding, section 3.3 states that proposed model is able to deal with more variants. And the original model is just a special case of this extension. I do not find there are extra complexity to deal with this extension. Why not combine section 3.2 and 3.3 into one.



**Q5 Detailed Comments To The Authors:**

It is better to reorgnize section 3.

**Q7 Justification For Your Score:**

Basically, this paper is well written and the technique seems sound and experiments valid the effect of proposed methods.



**Q9 Complying With Reviewing Instructions:**

1: Yes.

---

### Official Review · Reviewer_KPJp · 2022-04-15

**Q2(1) Originality/Novelty:** 2
**Q2(2) Significance/Impact:** 2
**Q2(3) Correctness/Technical Quality:** 3
**Q2(6) Clarity Of Writing:** 3
**Q6 Overall Score:** 4
**Q8 Confidence In Your Score:** 3

**Q1 Summary And Contributions:**

The work introduces a method to represent category, shape, and view factors from image data. The proposed approach is based on optimizing an ELBO objective. This work introduces some modifications to the model features by incorporating a better "inductive bias". The proposed method is able to achieve reasonable results on the datasets considered.

**Q2 Assessment Of The Paper:**

More detailed information regarding each of these aspects is given below:

**Q2(4) Quality Of Experiments (Optional):**

2: Fair: The experimental evaluation is weak: important baselines are missing, or the results do not adequately support the main claims.

**Q2(5) Reproducibility:**

2: Fair: Key resources (e.g., proofs, code, data) are unavailable but key details (e.g., proof sketches, experimental setup) are sufficiently well-described for an expert to confidently reproduce the main results.

**Q3 Main Strengths:**

1. Some experimental results are good.
2. The paper is in general easy to follow.


**Q4 Main Weakness:**

1. The proposed approach mainly uses an ELBO objective. The technical contribution is mainly on the empirical side in terms of how to incorporate the features. There are no theoretical insights, and the approach seems to be mainly hand-tuned based on heuristics.

2.  The training requires groups of images of the same object in multiple random views (as mentioned in section 4.1), it is unclear whether this approach can be applied to more common datasets where each object only has one view.

3. It would also be good to consider other stronger baselines (including GANs).

4. It would be good to consider more datasets, like CLEVR, Cars datasets, and even more challenging datasets where multiple random views of the same objects are not available.

5. The samples from swapping experiments (in fig5) are blurry and not of very high quality.

**Q5 Detailed Comments To The Authors:**


1. The proposed approach mainly uses an ELBO objective. The contribution is mainly on the empirical side in terms of how to incorporate the features. There are no theoretical insights, and the approach seems to be mainly hand-tuned based on heuristics.

2. Theorem 1 is not rigorous enough to be a theorem. There is also no rigorous proof. Perhaps it is better to call it an "observation".

3. The training requires groups of images of the same object in multiple random views (as mentioned in section 4.1). In many applications, multiple random views of the same object are not available. Would it be possible to apply this approach to commonly seen datasets where only one view of the same object is available?

4. It would be good to consider other stronger baselines. For instance, GAN-based approaches such as [1] do not require multiple views of the same object during training but could learn semantically meaningful features that can be used for clustering. Methods such as [2] can also be applied to perform swapping operations.

5. The results in figure 5 are blurry and not of very high quality.

[1] Niemeyer, Michael, and Andreas Geiger. "Giraffe: Representing scenes as compositional generative neural feature fields." Proceedings of the IEEE/CVF Conference on Computer Vision and Pattern Recognition. 2021.

[2] Karras, Tero, Samuli Laine, and Timo Aila. "A style-based generator architecture for generative adversarial networks." Proceedings of the IEEE/CVF conference on computer vision and pattern recognition. 2019.





**Q7 Justification For Your Score:**

The proposed work has limited theoretical novelty as the training objective is mainly an ELBO loss. Although the empirical results are reasonable, they are not very impressive. It would also be good to consider some stronger baselines and perform experiments on more datasets.

**Q9 Complying With Reviewing Instructions:**

1: Yes.

---

### Official Review · Reviewer_cYFn · 2022-04-15

**Q2(1) Originality/Novelty:** 3
**Q2(2) Significance/Impact:** 3
**Q2(3) Correctness/Technical Quality:** 3
**Q2(6) Clarity Of Writing:** 4
**Q6 Overall Score:** 7
**Q8 Confidence In Your Score:** 4

**Q1 Summary And Contributions:**

The paper proposes a deep generative model for jointly modeling and disentangling category, shape and viewpoint information using weak supervision, called CIGMO. The key idea is to learn multiple VAE architectures for the different categories of objects along with a categorization module using grouped data (multiple views of an object instance). The authors show their approach to be effective at unsupervised category discovery and one shot identification, and better than SOTA.

**Q2 Assessment Of The Paper:**

More detailed information regarding each of these aspects is given below:

**Q2(4) Quality Of Experiments (Optional):**

3: Good: The experimental evaluation is adequate, and the results convincingly support the main claims.

**Q2(5) Reproducibility:**

2: Fair: Key resources (e.g., proofs, code, data) are unavailable but key details (e.g., proof sketches, experimental setup) are sufficiently well-described for an expert to confidently reproduce the main results.

**Q3 Main Strengths:**

The paper proposes a novel method for disentangling categories, viewpoints and shape codes. This is a fairly under-studied and new problem. The proposed method is broadly applicable in AI and is not specific to a narrow problem domain and hence likely to be impactful.

The proposed method is technically sound.

The results are compelling and statistically significantly better than the existing SOTA method by large margins.

The paper is well written.

**Q4 Main Weakness:**

In the one-shot object recognition experiment with CIGMO, there is an assumption in that the category label c^{\hat} is known a priori. However, this does not appear to be the case for VAE or mixture of VAES. This can give CIGMO an unfair advantage.

Some details of the algorithm and architecture and training are not entirely clear, especially those that pertain to the connection between the categorizer and the specialized encoders. It would be helpful if the authors could provide code.

**Q5 Detailed Comments To The Authors:**

- "The model comprises multiple" --> "The model comprises of multiple"


**Q7 Justification For Your Score:**

Overall, the paper tackles a new problem, proposes a new method for it and shows it to work well in comparison to the existing SOTA methods in terms of various different metrics on two benchmark datasets. The method and experiments are adequate and technically sound. Hence I recommend accepting this paper.

**Q9 Complying With Reviewing Instructions:**

1: Yes.

---

### Official Review · Reviewer_vDed · 2022-04-15

**Q2(1) Originality/Novelty:** 2
**Q2(2) Significance/Impact:** 3
**Q2(3) Correctness/Technical Quality:** 2
**Q2(6) Clarity Of Writing:** 3
**Q6 Overall Score:** 6
**Q8 Confidence In Your Score:** 3

**Q1 Summary And Contributions:**

The motivation of the proposed paper is summarized in two aspects, i.e., 1) each category shape can be rendered in different views, and 2) shape diversities across categories are larger than within categories. The proposed method achieves the shape-view disentanglement through grouping the common shape and dismantling different views of an object based on GVAE. Both the quantitative and qualitative results demonstrate the effectiveness of this paper.

**Q2 Assessment Of The Paper:**

More detailed information regarding each of these aspects is given below:

**Q2(4) Quality Of Experiments (Optional):**

2: Fair: The experimental evaluation is weak: important baselines are missing, or the results do not adequately support the main claims.

**Q2(5) Reproducibility:**

3: Good: Key resources (e.g., proofs, code, data) are available and key details (e.g., proofs, experimental setup) are sufficiently well-described for competent researchers to confidently reproduce the main results.

**Q3 Main Strengths:**

1.	The proposed method is simple but effective, in terms of both the quantitative and qualitative results.
2.	CIGMO models VAE with the easy-to-get group category in a weakly supervised manner, and thus achieves better representation for clustering.
3.	The paper is well written and easy to follow.

**Q4 Main Weakness:**

1.     This paper is clearly an improvement based on GVAE. However, the network structure and principle comparison between these two methods are somewhat lacking. It looks like an explicit modeling with a group label added to GVAE.
2.	The visualization of generated images is not enough, especially lacking comparable references on the MVC Cloth dataset. Since the generated images are not realistic, comparative experiments are needed to prove the effectiveness of the method.
3.	In theory, this method can be directly applied to some larger models and datasets, but it seems no related experiments.

**Q5 Detailed Comments To The Authors:**

This paper demonstrates the good experimental results to confirm the proposed method. However, a few minor questions are listed as follows:
1)	It is not clear why the generation effect of MVC Cloth is relatively poor, whether it is because the model structure is not deep enough or the problem of the method itself.
2)	The author claims that the choice of group size K does not affect much the result. Please explain the reason or give some experimental results to prove it.

**Q7 Justification For Your Score:**

The proposed method is simple but effective, and achieve an improvement by a large margin in clustering accuracy. However, the method seems to be just a simple improvement of GVAE, which needs to be differentiated from more discussion on theory and network structure. Also, more experiments can be done to demonstrate effectiveness, such as larger models or datasets. Based on the above points, I suggest to rate “Weak accept”.

**Q9 Complying With Reviewing Instructions:**

1: Yes.

---

### Decision · Program_Chairs · 2022-05-15

**Decision:**

Accept (Poster)

**Comment:**

Meta Review: The proposed method aims to achieve category-shape-view disentanglement in images of objects by using a deep generative model which exploits grouping information in the data.

The reviewers appreciate the strong empirical results and acknowledge that this particular problem has not been studied extensively. The major concern raised is the somewhat limited novelty of the proposed methodology (a hand-designed ELBO objective).